# Subserosal Layer and/or Pancreatic Invasion Based on Anatomical Features as a Novel Prognostic Indicator in Patients with Distal Cholangiocarcinoma

**DOI:** 10.3390/diagnostics13223406

**Published:** 2023-11-09

**Authors:** Hisamichi Yoshii, Hideki Izumi, Rika Fujino, Makiko Kurata, Chie Inomoto, Tomoko Sugiyama, Toshio Nakagohri, Eiji Nomura, Masaya Mukai, Takuma Tajiri

**Affiliations:** 1Department of Gastroenterological Surgery, Tokai University Hachioji Hospital, Tokyo 192-0032, Japan; 2Department of Diagnostic Pathology, Tokai University Hachioji Hospital, Tokyo 192-0032, Japansugitomo@tokai-u.jp (T.S.); takumatajiri1003@yahoo.co.jp (T.T.); 3Department of Gastroenterological Surgery, Tokai University Hospital, Isehara 259-1193, Japan

**Keywords:** SS/Panc invasion, prognostic marker, AJCC 8th edition, T-staging system, depth of invasion, fibromuscular layer

## Abstract

The American Joint Committee on Cancer (AJCC) 8th edition T-staging system for distal cholangiocarcinoma (DCC) proposes classification according to the depth of invasion (DOI); nevertheless, DOI measurement is complex and irreproducible. This study focused on the fibromuscular layer and evaluated whether the presence or absence of penetrating fibromuscular invasion of DCC contributes to recurrence and prognosis. In total, 55 patients pathologically diagnosed with DCC who underwent surgical resection from 2002 to 2022 were clinicopathologically examined. Subserosal layer and/or pancreatic (SS/Panc) invasion, defined as penetration of the fibromuscular layer and invasion of the subserosal layer or pancreas by the cancer, was assessed with other clinicopathological prognostic factors to investigate recurrence and prognostic factors. According to the AJCC 8th edition, there were 11 T1, 28 T2, and 16 T3 cases, with 44 (80%) cases of SS/Panc invasion. The DOI was not significantly different for both recurrence and prognostic factors. In the multivariate analysis, only SS/Panc was identified as an independent factor for prognosis (hazard ratio: 16.1; 95% confidence interval: 2.1–118.8, *p* = 0.006). In conclusion, while the determination of DOI in DCC does not accurately reflect recurrence and prognosis, the presence of SS/Panc invasion may contribute to the T-staging system.

## 1. Introduction

Cholangiocarcinoma is anatomically divided into perihilar cholangiocarcinoma and distal cholangiocarcinoma (DCC) at the confluence of the three bile ducts, with DCC accounting for 29–42% of all biliary tract cancers [1]. On the one hand, the DCC region mainly extends from the confluence of the three bile ducts to the papillary bile duct region surrounded by the sphincter of Oddi, which passes through the pancreatic head region and penetrates the duodenal wall [2,3]. On the other hand, the histology of the bile duct wall is characterized by a three-layered structure that lacks the mucosal tendon plates and submucosa and tends to be thinner than the five-layered structure of the stomach and colon. Additionally, the area corresponding to the muscularis propria (i.e., the fibromuscular layer) is rich in fibrous connective tissues. The upstream and downstream areas also exhibit histological differences; in particular, the distal bile ducts upstream comprise loose muscle fibers (referred to as “scattered muscle fibers”), whereas the downstream areas possess thick fibromuscular layers and abundant bile duct accessory glands (mucous glands) in continuity with the bile duct epithelium [4].

The current T-classification system by the American Joint Committee on Cancer (AJCC) 8th edition adopts a format based on the distance of invasion of all cancers in the extrahepatic bile duct region [1,5,6,7]. Specifically, the depth of invasion (DOI), defined as the distance from the subepithelial basal lamina of the bile duct to the leading edge of invasion, has been reported to reflect prognosis [6]. Nevertheless, problems with the DOI measurement method, which is based on the importance of anatomic landmarks that classify staging according to the DOI (T1 < 5 mm; T2 5–12 mm; T3 > 12 mm), have been pointed out by pathologists. With the DOI measurement method, DCCs are subclassified into four categories according to the growth pattern of tumors, which is further complicated by the addition of cholangitis modification [6]. For instance, upstream of the distal bile duct, invasion into the subserosal layer is observed even in areas where the invasion distance is less than 5 mm because of thinning compared with that on the downstream side (Figure 1a). Furthermore, in DCC cases with a large tumor size, cancer cells have been often observed to have already spread to the outer serosa (black arrow in Figure 1b), which seems to be directly associated with the serosal surface involvement of cancer cells rather than the DOI (red arrow in Figure 1b) in terms of prognosis.

The distal bile duct is anatomically divided into two regions: (i) the region encased by the hepatoduodenal mesentery (extrapancreatic bile duct region) and (ii) the region adjacent to the pancreas (intrapancreatic bile duct region). It is questionable whether the T-classification based on the DOI alone is directly related to the prognosis of DCC, despite the anatomical and histological differences between the two (Figure 2). Thus, new prognosis-related T-factors must be identified, not DOI.

The present study focused on the anatomical and histological characteristics of distal bile ducts and compared the prognostic value of the extrapancreatic bile duct beyond the fibromuscular layer with that of other clinicopathological prognostic factors, such as the presence of subserosal layer invasion of the extrapancreatic bile duct and/or subserosal layer or pancreatic (SS/Panc) invasion through the intrapancreatic bile duct.

## 2. Materials and Methods

### 2.1. Study Population, Clinical Data, and Surgical Procedure, Outcome, and Follow-Up

This retrospective study analyzed the records of 55 patients who were clinically diagnosed with pancreatic head tumors (pancreatic head cancer, distal bile duct cancer, and duodenal papillary carcinoma) at the Tokai University Hachioji Hospital from April 2002 to October 2022 and were pathologically diagnosed with DCC after surgical resection. Patients who were pathologically diagnosed with intramucosal carcinoma (Tis) and died within 30 days after surgery were excluded from analysis. Pancreaticoduodenectomy and extrahepatic bile duct resection were performed in patients with surgically resectable DCC. Clinical data, including age, sex, tumor marker (carcinoembryonic antigen/carbohydrate antigen 19-9), adjuvant chemotherapy, and surgical procedures (surgical technique, blood loss, and operative time), were collected. In this study, the primary endpoints were (i) overall survival (OS), defined as the time from the date of surgery to the date of death, and (ii) survival, defined as the time from the date of surgery to the 5-year follow-up. Death (including death from other causes) was determined from the date of surgery to the date of death; however, death within 1 month after surgery was excluded. As a subanalysis, relapse-free survival (RFS) was calculated from the date of surgery to the date of recurrence on clinical images without local recurrence, distant metastasis, or secondary cancer. Follow-up evaluation was performed every 3 months for the first 2 years and every 6 months for 3–5 years using tumor markers and computed tomography images. The 5-year RFS and 5-year OS, as well as their respective medians and tracking rates, were calculated for all patients with DCC, and the prognosis of surgically resected DCC was evaluated.

This study was conducted in accordance with the principles embodied in the 1975 Declaration of Helsinki, as revised in 2013, and was approved by the Tokai University Ethics Committee (approval number: 23R078; date of approval: 17 August 2023).The requirement for informed consent from patients was waived owing to the retrospective nature of this study.

### 2.2. Pathological Data

After 10% formalin fixation, surgical specimens were sliced orthogonally to the bile ducts at approximately 5 mm intervals, and specimens were prepared with hematoxylin–eosin staining for pathological examination. The DOI was measured using Olympus cellSens software ver standard 4.1.1 and was measured based on four categories according to the growth pattern of tumors, as previously documented by Hong et al. [4]. Briefly, each patient was classified according to the AJCC 8th edition TNM classification. Clinicopathologic factors, pathological stage, tumor size, histologic type, SS/Panc invasion, lymphatic invasion (present/absent), perineural invasion (present/absent), venous invasion (present/absent), DOI, lymph node metastasis (present/absent), portal vein invasion (present/absent), artery invasion (present/absent), bile duct dissection plane (EM) (present/absent), hepatic bile duct transection (HM) (present/absent), and residual tumor status (R0 absent/R1 present) were evaluated by two pathologists (TT, TS). These factors were compared with those in T1/T2/T3 based on the AJCC 8th edition, and significant differences in clinicopathologic factors were examined. Discrepancies, if any, between the two pathologists and one surgeon were resolved by consensus. Univariate analysis of recurrence and prognosis was performed, and multivariate analysis including factors with *p* < 0.05 was conducted to identify factors for independent recurrence and prognosis.

### 2.3. Definition of SS/Panc Invasion

SS/Panc was reviewed on slides containing primary tumors with the most advanced areas of invasion. SS/Panc invasion was defined as invasion beyond the fibromuscular layer into the subserosal layer in the extrapancreatic bile duct region or beyond the fibromuscular layer into the subserosal layer or pancreas in the intrapancreatic bile duct region, or including both regions (Figure 2). Diagnosis was simultaneously evaluated by two pathologists (TT, TS) and one surgeon (HY) using the ONE microscopy method.

### 2.4. Statistical Analyses

All statistical analyses were performed using SPSS statistical software version 26 (SPSS Co., Ltd., Tokyo, Japan). Summary statistics were obtained using established methods and presented as percentages and medians. Nominal variables were analyzed using Pearson’s chi-squared test and Fisher’s exact test, whereas continuous variables were assessed using the U test. For the univariate analysis, Kaplan–Meier curves were constructed for the OS and RFS in each study group, and differences in survival were evaluated using the log-rank test. Independent recurrence and prognostic factors were analyzed using multivariate Cox regression analysis, with *p*-values of <0.05 indicating statistical significance.

## 3. Results

### 3.1. Clinicopathological Data on DCC

Table 1 summarizes the clinicopathological factors in 55 patients who underwent surgical resection for DCC. The study patients had a median age of 72 years (range: 45–85 years) and were predominantly male (73.6%). With respect to surgical procedures, pancreatoduodenectomy and extrahepatic bile duct resection were performed in 94.5% (*n* = 52) and 5.5% (*n* = 3) of patients, respectively. As for the pathological data, the AJCC 8th edition T-factors were T1 in 20% of patients (*n* = 11), T2 in 50.9% (*n* = 28), and T3 in 29.1% (*n* = 16); furthermore, the N-factors were N0 in 65.6% of patients (*n* = 36), N1 in 32.6% (*n* = 18), and N2 in 1.8% (*n* = 1), whereas pStaging was pStage I in 18.2% of patients (*n* = 10), pStage IIA in 41.8% (*n* = 23), pStage IIB in 38.2% (*n* = 21), and pStage IIIA in 1.8% (*n* = 1). Moderately differentiated adenocarcinoma was the most common histological type at 43.6% (*n* = 24). SS/Panc, lymphatic, perineural, and venous invasions were present in 80% (*n* = 44), 94.5% (*n* = 52), 83.6% (*n* = 46), and 45.5% (*n* = 25) of patients, respectively; a total of 79.2% (*n* = 42) of patients had R0. The median DOI was 8.53 mm (range: 1.0–22 mm). Among all patients with DCC, the 5-year RFS was 30%, with a median of 22.5 months at a follow-up rate of 92.7%, whereas the 5-year OS was 42%, with a median of 39.5 months at a follow-up rate of 87.2% (Figure 3).

### 3.2. Comparison of the 8th Edition AJCC T1–3 Clinicopathological Factors and Evaluation of RFS/OS

Table 2 presents the Kaplan–Meier curves for T1–3 in relation to T1–3. Lymph node metastasis and SS/Panc, lymphatic, venous, and portal vein invasions were significantly more common with a deeper DOI. T1 showed 63.6% (*n* = 7) in recurrence and 54.5% (*n* = 6) in death. For RFS, T1–3 did not stratify better (*p* = 0.129), and no significant differences were observed between T1 and T2 (*p* = 0.778) or between T2 and T3 (*p* = 0.095). Similarly, for OS, T1–3 were not stratified (*p* = 0.243), and no significant differences were observed between T1 and T2 (*p* = 0.703) or between T2 and T3 (*p* = 0.188) (Figure 4). T1 and T2 were inverted for both RFS and OS.

### 3.3. Recurrence and Prognosis of Clinicopathological Factors for DCC According to Univariate and Multivariate Analyses

The univariate analysis of RFS indicated significant differences in histological type (*p* = 0.010), venous invasion (*p* = 0.002), perineural invasion (*p* = 0.034), and SS/Panc invasion (*p* < 0.001). In the multivariate analysis, SS/Panc invasion was identified as an independent factor for relapse (hazard ratio (HR): 17.6; 95% confidence interval (CI): 2.3–129.8, *p* = 0.005) (Table 3). For RFS according to SS/Panc invasion, SS/Panc (−) showed a 5-year RFS of 89% (median not reached), whereas SS/Panc (+) was associated with a 5-year RFS of 16% (median: 12.0 months) (Figure 5).

The univariate analysis of OS also showed significant differences in venous invasion (*p* = 0.031), nerve invasion (*p* = 0.031), and SS/Panc invasion (*p* < 0.001). The multivariate analysis revealed that SS/Panc invasion was an independent prognostic factor (HR: 16.1; 95% CI: 2.1–118.8, *p* = 0.006) (Table 4). For OS, according to SS/Panc invasion, SS/Panc (−) showed a 5-year OS of 89% (median not reached), whereas SS/Panc (+) was associated with a 5-year OS of 29% (median: 23.8 months) (Figure 5).

## 4. Discussion

The present study investigated the measurement of infiltration distance in DCC based on the AJCC 8th edition and identified four problems. First, a uniform measurement of the DOI is difficult even with an accurate DOI measurement method because patients present with large tumor sizes and mixed growth patterns (papillary/nodular/invasive) [6,8]. Second, compared with the intrapancreatic bile duct region, the extrapancreatic bile duct region is not so thick in terms of bile duct wall thickness, not taking the anatomical differences in the distal bile duct into account. In our study, we also encountered a case of DCC in the extrapancreatic bile duct region with T1 invasion up to SS, with a poor prognosis (5-year RFS: 31%, 5-year OS: 34%), which does not reflect the stratification by T-factors in the AJCC 8th edition. Third, the AJCC T-staging system for DCC was created based on studies conducted by Hong et al. [4,6], which included all extrahepatic cholangiocarcinomas (including perihilar cholangiocarcinomas and DCCs treated by hepatic resection and pancreatoduodenectomy) and set the cutoff values for the DOI; hence, it is not a T-staging system specific to DCC [6,7]. In other words, the T-factors for DCC in the AJCC 8th edition do not sufficiently reflect recurrence and prognosis, and our opinion is similar to that of other case reports in the literature [9,10,11,12,13]. Fourth, clinical evaluation of the extent of cancer progression is difficult using preoperative imaging based on the 8th edition of the AJCC T-classification, and determining whether a patient should undergo upfront resection or chemotherapy is also challenging [8,12,13].

Lymph node metastasis, lymph node ratio, and residual tumor have been reported as factors influencing the prognosis of DCC [10,14,15,16]. In our study, no significant difference was noted in both RFS and OS for residual tumors. A certain consensus has not yet been reached because some reports in the literature have suggested that residual tumors have no effect on recurrence and prognosis, whereas other studies have shown that they contribute to prognosis [9,16,17,18]. As for lymph node metastasis, the relatively small number of cases is assumed to be a problem, given that lymph node metastasis significantly increases with the depth of T-factors. The present study did not reflect recurrence and prognosis; nevertheless, the univariate analysis of RFS showed a trend toward recurrence (*p* = 0.051).

The bile duct wall has a three-layered structure; however, the mucosal layer is often compromised due to conditions such as cholangitis or the presence of bile duct tubes, making detailed depth classification unreproducible. In our study, we encountered difficulties when trying to assess the AJCC T-class DOI, as cholangitis and preoperative biliary stenting drainage caused by cancer obstruction often resulted in mucosal injury or mucosal papillary proliferation. In contrast, it is usually easy for regular pathologists to evaluate the presence or absence of SS/Panc invasion in deeper layers with less inflammation. Additionally, in distal bile ducts, a two-group classification with the fibromuscular layer as the border may have prognostic relevance. In other words, in DCC, SS in the extrapancreatic bile duct region penetrating the fibromuscular layer may contribute to prognosis because of the presence of medium-to-large arteriovenous vessels, lymphatic vessels, and nerves distributed in the liver. Parenchymal pancreatic invasion by the intrapancreatic bile duct region that extends beyond the fibromuscular layer is also associated with the risk of pancreatitis. Pancreatic invasion is considered an invasion of other organs and increases the incidence of hematogenous metastasis; in fact, reported cases of prognostic factors have been confirmed in the literature [19,20,21,22]. Therefore, we defined SS/Panc as the penetration of cancer cells through the fibromuscular layer, regardless of whether it involves the internal and external pancreatic bile ducts. This parameter holds potential significance as a prognostic factor. The results indicated that the 5-year OS for DCC without SS/Panc invasion was 89%, whereas the 5-year OS for DCC with SS/Panc invasion beyond the fibromuscular layer was 29%, with a median of 23.8 months, which reflects an extremely poor prognosis. Notably, 28 of the 55 cases (50.9%) among the AJCC T1/2 patients exhibited SS/Panc invasion. Similarly, 100% of 5-year OS cases have been reported for gallbladder carcinoma in continuity with the bile ducts that do not extend beyond the muscularis propria/muscle layer, which corresponds to the fibromuscular layer. The invasion of the SS beyond the muscle layer is a similar poor prognostic factor, supporting our hypothesis in adjacent organs [17,23]. However, our study has some limitations, particularly its small sample size, potential for selection bias, and the inability to generalize the findings to different populations. Furthermore, this study has all the limitations and risks of bias inherent to cross-sectional studies.

Generally, the 5-year OS after surgical resection of DCC is reported to be 20–54% [24,25]. The 5-year OS in this study was 42% with a median of 39.5 months, which is almost comparable to that reported in the literature. No cases of T4 surgical resection were observed at our institution. The 5-year OS of DCC with concomitant portal vein resection is 0–15%, with a very poor prognosis [9,26,27,28]. There have also been few reports on arterial complications of resection in DCC, and the association with prognosis remains unclear [29]. Therefore, it may be reasonable to consider T4 as an unresectable factor. While neoadjuvant chemotherapy is currently not established for cholangiocarcinoma, it is likely that preoperative chemotherapy will be used for cholangiocarcinoma in the future, just as it is for pancreatic cancer. A randomized phase III trial (JCOG1920) with gemcitabine and cisplatin plus S-1 therapy is currently underway in Japan. Preoperative evaluation is not possible under the current T-factor convention for DCC, making it difficult to determine neoadjuvant chemotherapy [30,31]. However, if the depth of SS/Panc invasion is determined by considering the laminar structure, including the fibromuscular layer (i.e., SS/Panc invasion beyond the fibromuscular layer), preoperative diagnosis by intraductal ultrasonography/ultrasonic endoscopy/peroral cholangioscopy ultrasound or endoscopy may be possible [32,33,34].

In the future, biomarkers that may contribute to the further prognosis of SS/Panc invasion should be taken into account. Additionally, prospective randomized controlled trials on more aggressive neoadjuvant chemotherapy combined with preoperative molecularly targeted drugs and immune checkpoint inhibitors should be conducted to improve the prognosis of patients at a high risk for recurrence (poor prognosis group), in whom SS/Panc invasion is expected on imaging.

In conclusion, the presence or absence of SS/Panc invasion, which is a penetrating fibromuscular invasion, plays an important role as a recurrence and prognostic factor for DCC and may contribute to the T-staging system for DCC in the future.

## Figures and Tables

**Figure 1 diagnostics-13-03406-f001:**
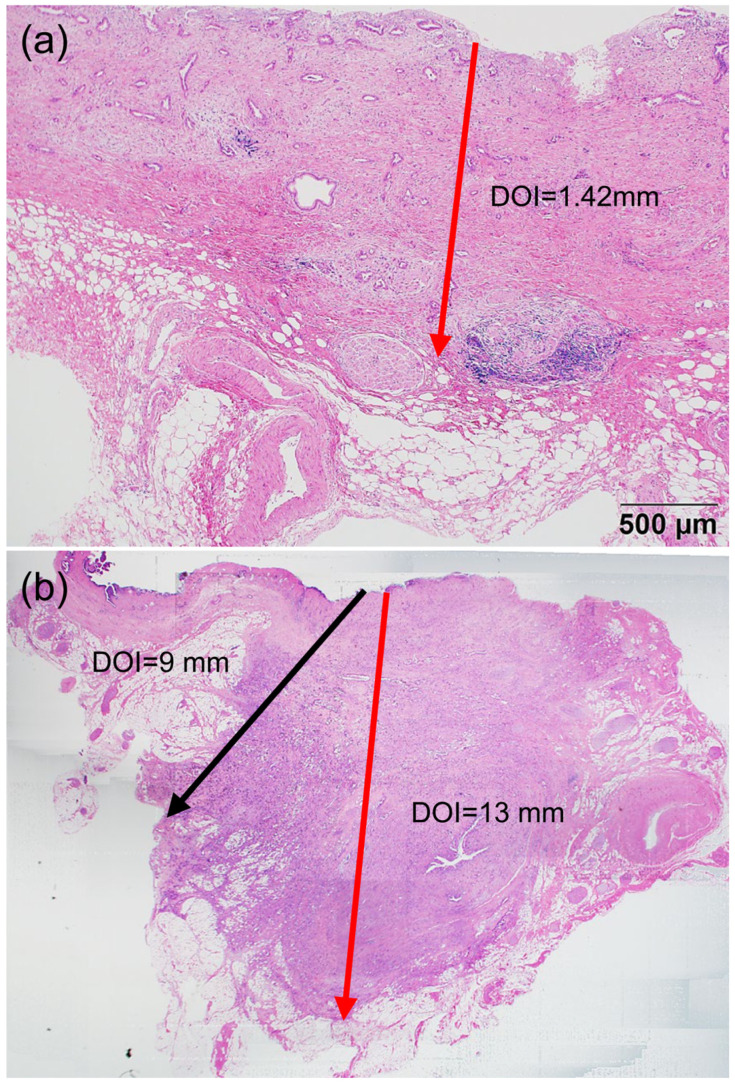
Problematic cases of DOI by the AJCC 8th edition T-staging system for DCC. (**a**) We examined the subserosal invasion arising from the extrapancreatic bile duct area in a 79-year-old male patient with DCC and found that, in addition to the recurrence of liver metastasis at 30 months after surgery, the patient died at 31 months. However, according to the current AJCC system, the DOI in the present case measured 1.42 mm (red arrow), which was classified as T1. Note the fibrous desmoplasia of cancer glands, complicated by the fibromuscular layer around the thin upstream wall of the distal bile ducts. (**b**) DCC in an 83-year-old male patient arising from upstream of the distal bile duct around the cystic duct junction. The DOI histologically measured 13 mm (red arrow) in loupe finding; however, the cancer cells had already spread to the serosal surface (black arrow) at 9 mm of the DOI from the lumen. The patient exhibited local recurrence at 5.6 months and died at 7.7 months after surgery. DOI, depth of invasion; DCC, distal cholangiocarcinoma; AJCC, American Joint Committee on Cancer.

**Figure 2 diagnostics-13-03406-f002:**
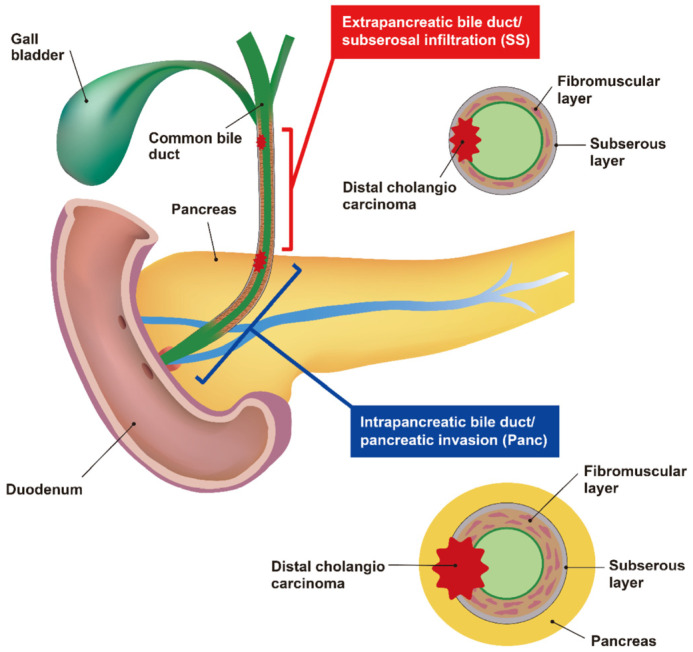
Anatomical distribution of the distal bile duct and definition of subserosal layer and/or pancreatic (SS/Panc) invasion. Based on the presence or absence of pancreatic tissues in the outer circumstance of the fibromuscular layer, the distal bile duct is mainly divided into upstream, which corresponds to the extrapancreatic bile duct, and downstream, which corresponds to the intrapancreatic bile duct. Histologically, the extrapancreatic duct comprises loose fibromuscular layers with a thin wall, whereas the intrapancreatic duct consists of dense fibromuscular layers with a thick wall. We defined SS/Panc invasion as cases in which the carcinoma progressed through the fibromuscular layer into invasion with subserosal layer or/and pancreatic tissues. Note the anatomical difference in the outer circumstance with subserosa or pancreas between the upstream or downstream distal bile ducts.

**Figure 3 diagnostics-13-03406-f003:**
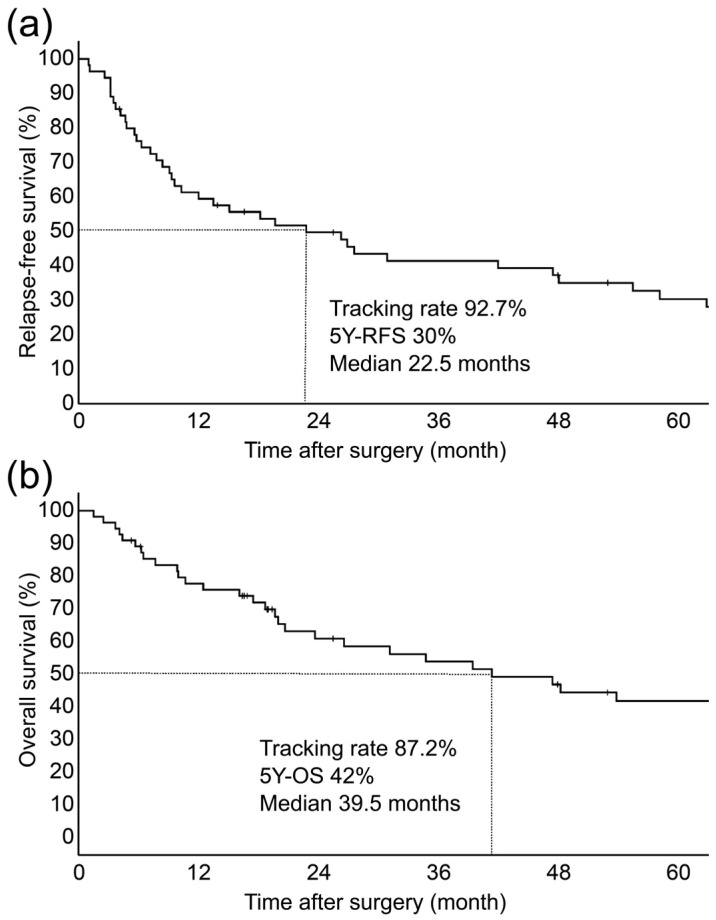
Kaplan–Meier curves for relapse-free survival (RFS) (**a**) and overall survival (OS) (**b**) in patients with DCC (*n* = 55). (**a**) The 5-year RFS for DCC was 30%, with a median of 22.5 months at a tracking rate of 92.7%. (**b**) The 5-year OS for DCC was 42%, with a median of 39.5 months at a tracking rate of 87.2%.

**Figure 4 diagnostics-13-03406-f004:**
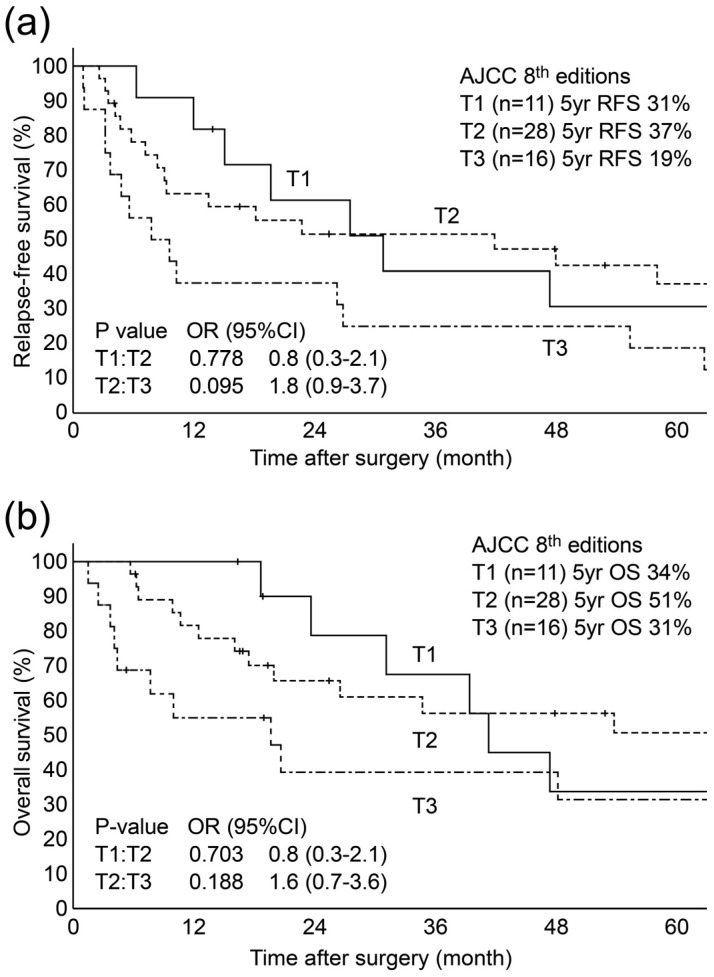
Kaplan–Meier curves for RFS (**a**) and OS (**b**) according to the AJCC 8th edition T-staging system. (**a**) Kaplan–Meier RFS in each T1–3 was not significantly stratified better (*p* = 0.129), and no significant differences between T1 and T2 (*p* = 0.778) or between T2 and T3 (*p* = 0.095) were observed. (**b**) Kaplan–Meier OS in each T1–3 was not significantly stratified (*p* = 0.243), and no significant differences between T1 and T2 (*p* = 0.703) or between T2 and T3 (*p* = 0.188) were found.

**Figure 5 diagnostics-13-03406-f005:**
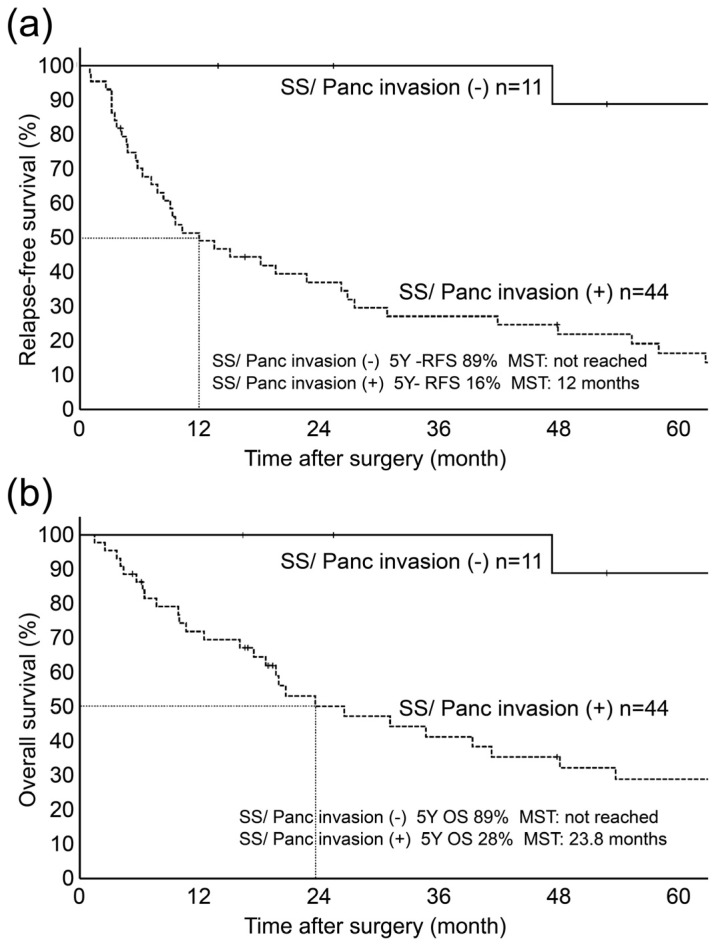
Kaplan–Meier curves for RFS (**a**) and OS (**b**) of DCC according to the presence or absence of SS/Panc invasion. (**a**) Kaplan–Meier RFS of outcome showed a significant difference between cases without SS/Panc invasion (5-year RFS: 89%) and those with SS/Panc invasion (5-year RFS: 16%) (*p* < 0.01, median: 23.8 months). (**b**) Kaplan–Meier OS of outcome showed a significant difference between cases without SS/Panc invasion (5-year OS: 89%, median not reached) and those with SS/Panc invasion (5-year OS: 29%, median: 23.8 months).

**Table 1 diagnostics-13-03406-t001:** Clinicopathological factors in relation to DCC.

Variables	*n* (%) or Median (Range)
Clinical information	
Age (years)	72 (45–85)
Sex: Male/Female	41 (73.6%)/14 (26.4%)
Carcinoembryonic antigen (ng/mL)	2.9 (0.6–16.2)
Carbohydrate antigen 19-9 (U/mL)	85.1 (1.0–22,000)
Adjuvant chemotherapy	15 (27.3%)
Recurrence	38 (69.1%)
Death	33 (60.0%)
Surgical procedure	
Pancreaticoduodenectomy	52 (94.5%)
Extrahepatic bile duct resection	3 (5.5%)
Blood loss (mL)	743 (162–3644)
Operation time (minutes)	279 (101–662)
Pathological information *	
T-stage	
T1	11 (20.0%)
T2	28 (50.9%)
T3	16 (29.1%)
N-factor	
N0	36 (65.6%)
N1	18 (32.6%)
N2	1 (1.8%)
pStage	
I	10 (18.2%)
II A	23 (41.8%)
II B	21 (38.2%)
III A	1 (1.8%)
Tumor size (mm)	24 (3–65)
Histological type	
Well	10 (18.1%)
Moderately	24 (43.6%)
Poorly	12 (21.9%)
Others	9 (16.4%)
SS/Panc invasion	44 (80.0%)
Lymphatic invasion	52 (94.5%)
Perineural invasion	46 (83.6%)
Venous invasion	25 (45.5%)
Depth of invasion (mm)	8.53 (1.01–22.0)
Lymph node metastasis	19 (34.5%)
Portal vein invasion	3 (5.5%)
Artery invasion	4 (7.3%)
EM	12 (21.8%)
HM	8 (14.5%)
R0/R1	42 (79.2%)/11 (20.8%)

* American Joint Committee on Cancer, 8th edition; DCC, distal cholangiocarcinoma; SS/Panc, subserosal layer and/or pancreatic. bile duct dissection plane (EM), hepatic bile duct transection (HM).

**Table 2 diagnostics-13-03406-t002:** Pathological factors and outcome according to the AJCC 8th edition T1–3 staging system.

Variables	T1 (*n* = 11)	T2 (*n* = 28)	T3 (*n* = 16)	*p*-Value
Age (years; median)	73(63–79)	72(45–84)	72(60–85)	0.627
Sex (Male:Female)	8:3	21:7	12:4	0.998
Tumor size (mm)	20 (3–55)	21 (4–65)	25 (10–50)	0.818
Histological type				
Well, Moderately, Pap	9 (81.8%)	19 (67.9%)	9 (56.3%)	0.378
Poorly, Others	2 (18.2%)	9 (32.1%)	7 (43.7%)	
SS/Panc invasion	6 (54.5%)	22 (78.6%)	16 (100%)	0.014
Lymphatic invasion	8 (72.7%)	28 (100%)	16 (100%)	0.002
Perineural invasion	8 (72.7%)	25 (89.2%)	13 (81.3%)	0.433
Venous invasion	2 (18.2%)	11 (39.2%)	12 (75.0%)	0.009
Depth of invasion (mm)	3.9 (1.0–5.0)	7.4 (5.0–1.1)	13 (12-22)	<0.001
Lymph node metastasis	1 (9.1%)	7 (25.9%)	11 (68.6%)	0.002
Portal vein invasion	0 (0%)	0 (0%)	3 (18.6%)	0.021
Artery invasion	0 (0%)	2 (7.1%)	2 (12.5%)	0.47
EM	1 (9.1%)	7 (25.0%)	4 (25.0%)	0.521
HM	2 (18.2%)	3 (10.7%)	3 (18.6%)	0.713
Residual tumor status R1	1 (9.1%)	6 (21.4%)	4 (25.0%)	0.576
Recurrence	7 (63.6%)	17 (60.7%)	14 (87.5%)	0.164
Death	6 (54.5%)	15 (53.6%)	12 (75.0%)	0.347

AJCC, American Joint Committee on Cancer. bile duct dissection plane (EM), hepatic bile duct transection (HM).

**Table 3 diagnostics-13-03406-t003:** Results of univariate (*p* < 0.05) and multivariate analyses of recurrence-free survival in relation to clinicopathological factors.

		*n*	Univariate Analysis	*p*-Value	Multivariate Analysis	*p*-Value
			Median (Range)		HR (95% CI)	
Age, years	≥75/<75	19/36	13.5 (0.9–26.0)/42.0 (6.0–77.9)	0.059		
Sex	Male/Female	41/14	22.8 (0–46.7)/10.4 (0–38.7)	0.302		
Histological	Tub1 + 2/por	37/18	42.0 (14.5–69.4)/4.8 (1.4–8.1)	0.010		
8th AJCC T-stage	T1/T2/T3	11/28/16	30.9 (14–47)/22 (0–86)/7 (0–15)	0.129		
Lymphatic invasion	Present/Absent	52/3	22.8 (4.9–40.6)/-	0.306		
Perineural invasion	Present/Absent	46/9	18.2 (4.4–31.9)/-	0.034		
Venous invasion	Present/Absent	25/30	7.8 (2.6–12.9)/48.1 (17.6–78.5)	0.002	2.3 (1.1–4.4)	0.012
LN metastasis	Present/Absent	19/36	10.3 (0–15.7)/42 (14.0–69.9)	0.051		
SS/Panc invasion	Present/Absent	44/11	12.0 (4.5–319.4)/-	<0.001	17.6 (2.3–129.8)	0.005
EM	Present/Absent	43/12	9.1 (0.4–17.7)/26.9 (12.2–41.5)	0.183		
HM	Present/Absent	47/8	7.8 (0–17.7)/26.9 (11.5–42.2)	0.181		
Residual tumor status	R0/R1	44/11	26.9 (13.8–39.9)/12.0 (2.6–21.3)	0.118		

HR, hazard ratio; CI, confidence interval; LN, lymph node. bile duct dissection plane (EM), hepatic bile duct transection (HM).

**Table 4 diagnostics-13-03406-t004:** Results of univariate (*p* < 0.05) and multivariate analyses of overall survival in relation to clinicopathological factors.

		*n*	Univariate Analysis	*p*-Value	Multivariate Analysis	*p*-Value
			Median (Range)		HR * (95% CI)	
Age, years	≥75/<75	19/36	34.8 (4.4–65.1)/47.5 (12.4–82.5)	0.214		
Sex	Male/Female	41/14	41.4 (0–86.8)/39.5 (0–86.1)	0.363		
Histological	Tub1 + 2/por	37/18	41.4 (15.2–67.5)/34.8 (0–82.7)	0.556		
AJCC 8th edition T-stage	T1/T2/T3	11/28/16	41.4 (35–46)/73 (10–136)/19 (2–37)	0.243		
Lymphatic invasion	Present/Absent	52/3	39.5 (13.8–65.1)/-	0.318		
Perineural invasion	Present/Absent	46/9	34.8 (14.0–55.5)/-	0.031		
Venous invasion	Present/Absent	25/30	19.7 (13.7–25.6)/73.5 (25.5–121.4)	0.031		
LN metastasis	Present/Absent	19/36	20.7 (11.5–29.8)/53.9 (13.6–94.1)	0.227		
SS/Panc invasion	Present/Absent	44/11	26.0 (11.4–41.7)/-	<0.001	16.1 (2.1–118.8)	0.006
EM	Present/Absent	43/12	48.3 (1.8–94.7)/41.4 (19.1–63.6)	0.622		
HM	Present/Absent	47/8	5.1 (13.5–33.8)/48.3 (22.1–74.4)	0.268		
Residual tumor status	R0/R1	44/11	41.4 (23.0–59.7)/23.7 (0–103)	0.443		

* HR, hazard ratio; CI, confidence interval; LN, lymph node. bile duct dissection plane (EM), hepatic bile duct transection (HM).

## Data Availability

The data presented in this study are available on request from the corresponding author. The data are not publicly available because of ethical concerns (informed consent not obtained).

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
