# Peer review of "Subserosal Layer and/or Pancreatic Invasion Based on Anatomical Features as a Novel Prognostic Indicator in Patients with Distal Cholangiocarcinoma"

_diagnostics, 2023, doi:10.3390/diagnostics13223406_

Round 1

Reviewer 1 Report

Comments and Suggestions for Authors

As the authors advocated, the current 8th AJCC staging of distal cholangiocarcinoma holds several concerns and this study clearly showed the issues to be addressed.  Probably the AJCC T category will stratify patients' prognosis with more increased number of patents. I think it would be better to emphasize how the AJCC T category may inappropriately estimate ss/panc invasion as was presented in Fig. 1 in the introduction. 

1. Were there any factors that was associated with positive ss/panc invasion in the patients with AJCC T1/T2 or negative ss/panc invasion in those with T3 disease?

2. Were there any influence of preoperative biliary drainage or cholangitis on improper evaluation of AJCC T class with standard reference as ss/panc invasion?

3. The authors described that two pathologists evaluated the pathologic staging. Can the authors provide Kappa coefficient value for AJCC T staging or ss/panc invasion between them?

4. In Fig. 5b, delete the words "Time after surgery (months)" on the survival curve.

Reviewer 2 Report

Comments and Suggestions for Authors

In the manuscript titled "Subserosal Layer and/or Pancreatic (SS/Panc) Invasion Based on Anatomical Features as a Novel Prognostic Indicator in Patients with Distal Cholangiocarcinoma," Hisamichi Yoshii and colleagues describe their findings from a retrospective study on the AJCC 8th edition T-staging system for distal cholangiocarcinoma. It is a very relevant clinical topic. Overall, the manuscript is well written. However, some points need to be addressed.

Major points:

The authors' definition of subserosal layer or pancreatic (SS/Panc) invasion can be found under 2.3 Definition of SS/Panc Invasion and can be found in the discussion in line 274 (“we defined depth of penetration through the fibromuscular layer”). However, it remains unclear if the measurement remains a categorical variable (invasion yes/no, as it was used in uni- and multivariate analysis) OR a continuous variable measuring the actual depth and then categorizing different groups (such as the T-staging). If the latter is true, then it remains unclear where the measurement starts from. Does it start from the fibromuscular layer rather than the mucosa? Or is it a combination of both: yes/no and additional infiltration depth? Please comment on that. I would suggest precisely clarifying this matter in a graphic that could be added to Figure 2 (or adjust Figure 2). In general, I assume it makes sense to categorize it as yes/no to find a unique stratification factor for intrapancreatic and extrapancreatic DCCs.

In Table 2, the authors grouped the patients according to their T-status. No difference in OS and RFS could be detected. It would be interesting to see if other confounding factors were equal between the groups (age, sex, etc.). If there are no differences, it does not need to be shown in the table but mentioned in the text. If there are differences, they should be considered and discussed.

It would be very interesting for the reader to additionally see Kaplan-Meier curves for cancer-specific survival.

Minor points:

Please check the formatting of Figure 5b; "Time after surgery (month)" is covering the graph.

Lines 272-275 are hard to read; consider splitting them into two sentences.

Comments on the Quality of English Language

The quality is fine and only minor grammar/syntax errors need to be corrected. The overall structure is very good and makes it easy to read.

Round 2

Reviewer 1 Report

Comments and Suggestions for Authors

The authors adequately responded to each reviewer's comments. I have no additional comments for further review.

Reviewer 2 Report

Comments and Suggestions for Authors

Thanks for addressing the issues.

Comments on the Quality of English Language

No comments.